# Gut microbiome mediates the associations between lifestyle factors and risk of colorectal high-risk adenoma: results from a population-based cohort study

Kai Song,[1,2] Jiahui Luo,[1,3] Yuhan Zhang,[4] Dong Wu,[2] Hongda Chen,[1] Min Dai[3]

**ABSTRACT**    Lifestyle factors exert influence on the risk of colorectal cancer (CRC) and its precursors. However, the interaction between gut microbiota and lifestyle factors concerning colorectal high-risk adenomas (HRAs), as well as the specific microbial taxa implicated, remains underexplored. Here, we aimed to investigate the impact of common lifestyle factors on HRAs and to explore the potential mediating roles of gut microbiota in these effects. A total of 3,827 participants were enrolled from our multicenter CRC screening cohort. Lifestyle factors over the 12 months preceding enrollment were assessed via questionnaires. Fecal samples were collected upon enrollment and analyzed using 16S rRNA sequencing. Multivariate analyses were used to identify lifestyle-related risk factors for HRA, followed by the application of the multivariate association with linear models (MaAsLin2) to explore associations between microbiota and lifestyle risk factors, with causal mediation analysis employed to evaluate the gut microbiota's mediating effect between lifestyle factors and HRA risk. A total of 272 patients with HRA and 1,253 controls were included. Independent risk factors for HRA were identified as high body mass index, number of pack-years smoked >30, alcohol consumption >4 units/week. These lifestyle factors were significantly associated with the relative abundance of various microbial genera. Notably, genus *Fusobacterium* and *Tyzzerella 4* were found to partially mediate the increased risk of HRA due to alcohol and cigarette consumption, respectively. These findings provide new insights for microbiota-targeted interventions or lifestyle-based prevention strategies to reduce the risk of CRC, offering a novel and actionable approach to early prevention.

**IMPORTANCE** Lifestyle choices, such as diet, smoking, and alcohol consumption, are known to influence colorectal cancer risk, but the role of gut bacteria in mediating this process remains underestimated. To address this gap, our study aimed to explore the connections between lifestyle factors, gut microbes, and colorectal precancerous growths, referred to as high-risk adenomas (HRAs). We observed a dual association whereby obesity, heavy smoking, and excessive alcohol consumption were linked to both an increased risk of HRAs and distinct changes in gut bacteria. Importantly, smoking and alcohol consumption are associated with increased cancer risk, in part, through certain bacteria such as genus *Fusobacterium* and *Tyzzerella 4*. These findings reveal how gut microbes may act as a hidden bridge between lifestyle and disease development. Our discovery of these microbial mediators reveals novel opportunities for HRA prevention through lifestyle modifications or probiotic interventions targeting this carcinogenic pathway prior to malignant transformation.

**CLINICAL TRIALS** This study is registered with the Chinese Clinical Trial Registry as ChiCTR1800015506.

**Peer Reviewers** Jinshui Yang, China Agricultural University, Beijing, China; Wei-Hua Chen, Huazhong University of Science and Technology, Wuhan, China; Zhihua Liu, Tsinghua University, Beijing, China

Address correspondence to Hongda Chen, chenhongda@pumch.cn, or Dong Wu, wudong@pumch.cn.

The authors declare no conflict of interest.

**KEYWORDS** colorectum, precancerous lesions, lifestyle, gut microbiota, mediation

Colorectal cancer (CRC) was the third most frequently diagnosed cancer worldwide in 2022, accounting for 9.6% of all new cancer cases and causing a significant health and socioeconomic burden (1). In China, CRC is the second most common cancer and the fourth leading cause of cancer mortality (2). Prevention and early intervention targeting precancerous lesions is a more cost-effective strategy for reducing the incidence and mortality of CRC (3, 4). Colorectal high-risk adenomas (HRAs) are a category of precancerous lesions defined based on clinical research evidence over the past decade, including advanced adenoma and cases with three or more adenomas (5). Despite resection, these lesions remain at high risk for both recurrence and malignant transformation to CRC, indicating inherent aggressive potential (5, 6). Therefore, understanding the risk factors and underlying mechanisms contributing to HRA development is essential for improving early prevention strategies.

While multiple studies in CRC have shown that unhealthy lifestyle factors, including excessive intake of red meat (7, 8), smoking (9), and alcohol consumption (9), increase disease risk, evidence clarifying their contribution to HRA remains scarce. Moreover, the underlying mechanism through which these external exposures promote neoplastic transformation at the precancerous stage is not fully understood.

The gut microbiota, as a key component of the intestinal microecosystem interacting with the gut epithelial mucosa, is closely linked to colorectal adenoma and CRC development through compositional alterations (10–13). These alterations, including the gut microbiota-derived metabolites that induce chronic inflammation and immune dysfunction, as well as elevated serum lipopolysaccharide levels (14), suggest that the gut microbiota may mediate the impact of adverse lifestyle factors on colorectal neoplastic risk, warranting further investigation. However, while this overall mediating role is plausible, evidence at the taxonomic level remains scarce. Specifically, the gut microbial taxa that interact with particular lifestyle factors to modulate HRA risk have yet to be clearly identified.

To address this gap, we conducted a large, multicenter cohort study recruiting participants undergoing CRC screening across China. Our aim was to identify lifestyle-related risk factors for HRA, evaluate their associations with gut microbiota, and identify certain bacterial taxa mediating these relationships. This study provides a novel perspective on the microbial mechanisms linking lifestyle and early colorectal neoplasia, contributing to establishing microbiota-based or lifestyle-targeted strategies in preventing CRC from its precancerous stage.

## MATERIALS AND METHODS

### Study participants

This study was based on the multicenter cohort study TARGET-C, initiated in May 2018 (registry number ChiCTR1800015506) (15, 16). In short, the main objective of TARGET-C was to compare the effectiveness of colonoscopy-based fecal immunochemical tests (FIT) and risk-adapted screening strategies for CRC screening in China. Participants were required to undergo colonoscopy and collect stool samples within 24 h prior to bowel preparation. Epidemiological data (e.g., smoking and alcohol consumption) and clinical data (e.g., polyp pathology, size, number) were also collected. The study received approval from the Ethics Committee of the National Cancer Center/Cancer Hospital, Chinese Academy of Medical Sciences, and Peking Union Medical College (18-013/1615).

For the present study, screening patients aged 50–74 with informed consent were enrolled. The inclusion criteria comprised individuals with no abnormalities detected at baseline colonoscopy (healthy controls, HCs) and patients diagnosed with low-risk adenomas (LRAs), HRA, or CRC. Additionally, stool samples were required for microbiome sequencing. Exclusion criteria, in addition to the ineligible subjects as per the study protocol, included missing data, lack of corresponding stool samples, or poor-quality 16S

rRNA sequencing. For more details on patient selection, please refer to Fig. S1. HRA were defined as adenomas ≥ 10 mm, those with tubulovillous or villous histology, adenomas with high-grade dysplasia, or ≥3 adenomas (5), to focus on the relationship between lifestyle factors and high-risk precancerous lesions. LRAs were defined as colorectal adenomas without the above characteristics.

## Epidemiological data collection

After participants were enrolled in the cohort, trained researchers were responsible for collecting epidemiological data, including age, gender, body mass index (BMI), family history of CRC in first-degree relatives, history of colorectal polyps, number of pack-years smoked, average alcohol consumption, and use of non-steroidal anti-inflammatory drugs (NSAIDs) over the past 12 months. Average alcohol consumption was expressed in terms of how many units of alcohol were consumed per week. We defined one unit of alcohol (one drink) as 350 mL of beer, 150 mL of wine, or 50 mL of baijiu (a distilled spirit typically consumed in 50 mL servings in China) (17, 18). When consuming two or more types of alcohol, the alcohol consumption is cumulative based on these definitions.

## Stool sample collection and 16S rRNA gene amplicon sequencing

Stool samples collected from the patients were frozen at −20°C until DNA extraction. DNA was extracted using the QIAamp Fast DNA Stool Mini Kit. 16S rRNA gene amplicon sequencing was performed. The V4 region of the microbial 16S rRNA gene was amplified and sequenced on the Illumina MiSeq sequencing platform (Illumina, San Diego, CA). To minimize end-reads sequencing errors, all reads were truncated at the 150th base with a median Q score of >20. Noise sequences, chimeras, and singletons were removed, and amplicon sequence variants (ASVs) were inferred from the clean sequencing reads using the DADA2 pipeline integrated into Qiime2 (19). Taxonomy was assigned to each ASV using the classify-sklearn method with the q2-feature-classifier plugin built on the Greengenes database (version 13.8). To quantify taxonomic composition, all sequences were rarefied to a uniform sampling depth of 10,000. Additionally, only taxa present in at least 1% of the samples with an average relative abundance greater than 0.01% were included in the downstream analyses.

## Statistical analysis

Continuous variables were presented as mean and standard deviation, while categorical variables were shown as frequency and proportion. Univariate analysis followed by multiple covariate adjustments was conducted to assess independent risk factors for HRA. Statistical analyses were conducted using R software (v.4.3.1). The differences in categorical variables were evaluated using the Chi-squared test, while the Kruskal-Wallis $H$ test and Wilcoxon test were applied to examine differences in continuous variables. The cut-off value of 30 pack-years for smoking history was based on previous studies (20, 21), while the cut-off value for BMI was determined according to the established thresholds for overweight (BMI 24–28 kg/m$^2$) and obesity (BMI ≥28 kg/m$^2$) in the Chinese population (22).

For the 16S rRNA data, microbial diversity was assessed through α-diversity using the Shannon index, and β-diversity was calculated using the Bray-Curtis distance. These analyses were conducted using the "vegan" package in R. The associations between lifestyle factors and β-diversity dissimilarities were analyzed through permutational multivariate analysis of variance (PERMANOVA, 999 permutations), adjusted for covariates including age, gender, BMI, family history of CRC, the recruitment region of participants and the diagnosis (to adjust for the effects of disease on gut microbiota) (23, 24). To analyze the association between gut microbiota and lifestyle factors, we used the multivariate association with linear models (MaAsLin2) method, implemented via the R package "MaAsLin" (25). We log-transformed the relative abundances of microbial features and standardized lifestyle factors into $Z$-scores before including them in the

MaAsLin2 models. Multiple hypothesis testing was addressed by controlling the false discovery rate (FDR < 0.25) via the Benjamini-Hochberg method, a statistically significant threshold standard in microbiome research (25–28). The MaAsLin2 model was adjusted for age, gender, BMI, family history of CRC, recruitment region, use of NSAIDs, history of colorectal polyp, and disease status. In addition to conducting correlation analysis in our own data set, an external cohort of the Japanese population was used to validate the association between lifestyle factors and gut microbial genera (29) (for details about the external cohort, see supplementary material).

In the differential analysis of gut microbiota, microbial taxa with a $P$ value < 0.05 and a log2 fold-change ≥1.0 were considered significant. Additionally, taxa identified by the linear discriminant analysis effect size (LEfSe) method were included in downstream analyses.

Causal mediation analysis was conducted using the "mediation" R package, to test the hypothesis that the observed effect of lifestyle factors on the risk of HRA is mediated by its effect on lifestyle factors-associated microbial genus. The direct and indirect effects were calculated using a bootstrap with 1,000 simulations. Coefficient estimates, 95% confidence intervals, and the $P$ values were reported.

Sensitivity analyses were performed on different clinical endpoint (advanced adenomas). Subgroup analyses were performed among participants of different genders.

## RESULTS

### Participant characteristics

A total of 1,575 participants with qualified sequencing data were included (Fig. S1, recruitment regions are shown in Fig. 1A). Following baseline colonoscopy and exclusion of participants with missing lifestyle data, the cohort comprised: CRC (24 patients), HRA (272 patients), LRA (465 patients), and healthy controls (HC, 788 patients). In univariate analysis, comparing the HRA group with the control group (which includes LRA and HC), the HRA group had a higher mean age (61.9 vs 60.5), a larger proportion of males (76.1% vs 54.0%), and a higher mean BMI (24.7 vs 24.1) (Table 1). In terms of lifestyle factors, the HRA group exhibited a higher number of pack-years smoked ($P$ < 0.001) and greater alcohol consumption ($P$ < 0.001).

### Association between lifestyle factors and the risk of high-risk adenoma

After adjusting for multiple covariates (age, gender, BMI, education level, recruitment region of participants, family history of CRC in first-degree relatives, history of colorectal polyps), the independent risk factors for HRA were limited to number of pack-years smoked >30 (adjusted OR: 1.44, 95% CI: 1.00–2.06, $P$ = 0.048), alcohol consumption >4 units/week (adjusted OR: 1.65, 95% CI: 1.09–2.48, $P$ = 0.017), and BMI (adjusted OR: 1.06, 95% CI: 1.01–1.11, $P$ = 0.016) (Table 2). Given that lifestyle factors can influence the gut microbiota, we further sought to explore how these lifestyle risk factors for HRA affect the gut microbiome.

### Association between lifestyle factors and gut microbiota

We first explored the effects of the aforementioned lifestyle risk factors on gut microbiota diversity. Significant differences in the Shannon index were observed among patients with different BMI categories, while no significant differences were found with respect to cigarette consumption, alcohol consumption, or richness (Fig. 1B through G). However, after adjusted for covariates (age, gender, recruitment region of participants, BMI, family history of CRC in first-degree relatives, history of colorectal polyp, use of NSAIDs, disease status), the β diversity based on the Bray-Curtis distance matrix among different BMI categories (PERMANOVA adjusted $P$ = 0.001), cigarette consumption (PERMANOVA adjusted $P$ = 0.001), alcohol consumption (PERMANOVA adjusted $P$ = 0.001) showed significant differences (Fig. 1H through J). Therefore, lifestyle factors can influence the composition of the gut microbiota.

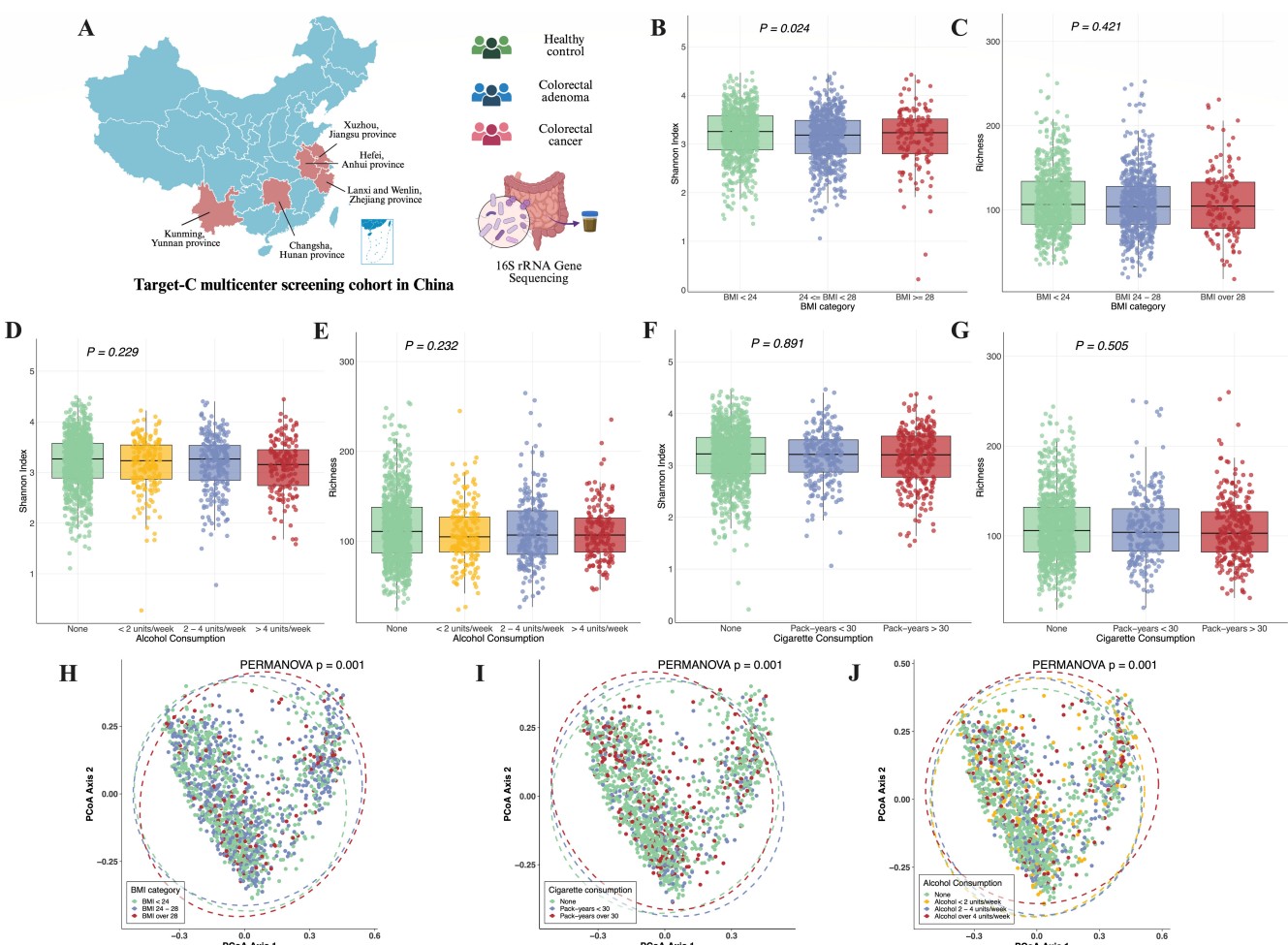

**FIG 1** The impact of lifestyle risk factors on gut microbiome. (A) Schematic diagram showing the geographic locations of each recruitment region. (B) Effect of BMI categories on Shannon index of gut microbiota. (C) Effect of BMI categories on richness of gut microbiota. (D) Effect of alcohol consumption on shannon index of gut microbiota. (E) Effect of alcohol consumption on richness of gut microbiota. (F) Effect of pack-years of smoking on Shannon index of gut microbiota. (G) Effect of pack-years of smoking on richness of gut microbiota. (H) Effect of BMI on β-diversity of gut microbiota. The principal component analysis (PCoA) plots were constructed based on the Bray-Curtis distance matrix. The *P* value on the figure represents the result of the covariate-adjusted permutational multivariate analysis of variance (PERMANOVA). (I) Effect of pack-years of smoking on β-diversity of gut microbiota. (J) Effect of average alcohol consumption on β-diversity of gut microbiota.

Further analysis using the MaAsLin2 method revealed that several gut genera were associated with lifestyle risk factors (Fig. 2A). After adjusted for covariates (age, gender, recruitment region of participants, BMI, family history of CRC in first-degree relatives, history of colorectal polyp, use of NSAIDs, disease status) and using a cutoff of FDR < 0.25, the genera associated with lifestyle risk factors are listed in Table S1. Specifically, genera positively correlated with alcohol consumption included *Fusobacterium* (coefficient = 0.285, FDR = 0.080), *Negativibacillus* (coefficient = 0.204, FDR = 0.017), and *Prevotella 9* (coefficient = 0.533, FDR = 0.060), while *Bifidobacterium* (coefficient = −0.297, FDR = 0.080) negatively correlated with alcohol consumption. Genera positively correlated with number of pack-years smoked included *Actinomyces* (coefficient = 0.266, FDR < 0.001).

We further conducted MaAsLin2 analysis, adjusted for age, gender, BMI, alcohol consumption, Brinkman index of smoking, and disease status (CRC or control), in an external cohort consisting of 299 CRC cases and 299 controls after 1:1 nearest matching by age, gender, and BMI (characteristics of the study population are shown in Table S2) (29). The associations between gut microbiota and lifestyle factors were validated, and

**TABLE 1** Characteristics of the study population[a]

| Variable | Control (n = 1,253) | HRA (n = 272) | P value[b] |
|---|---|---|---|
| Age, mean (SD) | 60.5 (6.3) | 61.9 (6.2) | **<0.001** |
| Female/male | 577/676 | 65/207 | **<0.001** |
| BMI (kg/m$^2$), mean (SD) | 24.1 (2.8) | 24.7 (3.0) | **<0.001** |
| Education level, n (%) | | | 0.608 |
| <High school | 936 (74.7%) | 210 (77.2%) | |
| High school and equivalent | 225 (18.0%) | 42 (15.4%) | |
| Postsecondary graduate | 92 (7.3%) | 20 (7.4%) | |
| Family history of CRC in first-degree relatives, n (%) | 142 (11.3%) | 38 (14.0%) | 0.263 |
| History of colorectal polyps, n (%) | 36 (2.9%) | 5 (1.8%) | 0.477 |
| Use of NSAIDs, n (%) | | | 0.734 |
| No | 1,194 (95.3%) | 257 (94.5%) | |
| Sometimes | 27 (2.2%) | 8 (2.9%) | |
| Regular | 32 (2.6%) | 7 (2.6%) | |
| Number of pack-years smoked, n (%) | | | **<0.001** |
| None | 870 (69.4%) | 148 (60.3%) | |
| <30 | 185 (14.8%) | 47 (17.3%) | |
| >30 | 198 (15.8%) | 77 (28.3%) | |
| Alcohol consumption, n (%) | | | **<0.001** |
| None | 791 (63.1%) | 138 (50.7%) | |
| <2 unit/w | 156 (12.5%) | 26 (9.6%) | |
| 2–4 unit/w | 195 (15.6%) | 60 (22.1%) | |
| >4 unit/w | 111 (8.9%) | 48 (17.6%) | |

[a]BMI, body mass index; CRC, colorectal cancer; HRA, high-risk adenoma; NSAID, nonsteroidal anti-inflammatory drug; SD, standard deviation.
[b]Bold values indicate statistically significant difference.

we observed a consistency in the relationships between lifestyle factors and gut genera (e.g., *Bifidobacterium* and alcohol consumption, *Hungatella* and BMI, *Allisonella* and BMI) (Fig. 2B; Table S3). Notably, the genus *Fusobacterium* was also positively correlated with alcohol consumption in this external cohort (coefficient = 0.638, P value = 0.018, FDR = 0.200), similar with the findings in our cohort.

## Interaction between gut microbiota and lifestyle factors promotes high-risk adenoma

Next, we aimed to explore whether lifestyle-related microbiota play a role in increasing the risk of HRA. We began by analyzing the differential microbiota between HRA and control groups. Differential analysis identified three microbial genera that were significantly upregulated and six that were significantly downregulated in HRA patients compared to healthy controls, with a P value < 0.05 and a log$_2$ fold-change ≥1.0 (Fig. 3A). Notably, the relative abundance of genus *Fusobacterium* was upregulated in the HRA group.

LEfSe analysis identified 15 genera, which were differentially abundant in the two groups (Fig. 3B). In the HRA group, there was an enrichment of the phyla *Fusobacteria*, *Firmicutes*, and *Actinobacteria*, including the genera *Fusobacterium*, *Tyzzerella 4*, *Phascolarctobacterium*, *Gemella*, *Ruminococcaceae UCG-004*, *Actinomyces*, and *Terrisporobacter*. In contrast, the genera *Oscillibacter*, *Lachnospiraceae UCG-004*, *Lachnospira*, *Parasutterella*, *Dialister*, *Faecalibacterium*, *Prevotellaceae NK3B31 group*, and *Ruminococcaceae UCG-003* were more abundant in the control group.

Integrating findings from both MaAsLin 2 and LEfSe analyses revealed significant overlap between genera differentially abundant in HRA vs controls and those associated with lifestyle factors. These overlapping genera include *Tyzzerella 4*, *Fusobacterium*, *Actinomyces*, and *Oscillibacter*. In Spearman correlation analysis, we found significant correlations between lifestyle-associated differential genera and adenoma

**TABLE 2** Association between lifestyle factors and risk of high-risk adenoma adjusted for different covariates[a]

| Lifestyle factors | Model 1[b] | | | Model 2[c] | | | Model 3[d] | | |
|---|---|---|---|---|---|---|---|---|---|
| | Coefficient | OR (95% CI) | P value[e] | Coefficient | OR (95% CI) | P value[e] | Coefficient | OR (95% CI) | P value[e] |
| BMI (kg/m$^2$) (continuous variable) | 0.07 | 1.07 (1.02–1.12) | **0.008** | 0.06 | 1.07 (1.02–1.12) | **0.008** | 0.06 | 1.06 (1.01–1.11) | **0.016** |
| BMI category (kg/m$^2$) (categorical variable) | | | | | | | | | |
| BMI < 24 | REF | | | REF | | | REF | | |
| BMI 24–28 | 0.21 | 1.23 (0.92–1.64) | 0.156 | 0.20 | 1.22 (0.92–1.63) | 0.164 | 0.20 | 1.22 (0.92–1.63) | 0.174 |
| BMI ≥ 28 | 0.54 | 1.71 (1.08–2.65) | **0.019** | 0.54 | 1.71 (1.09–2.66) | **0.018** | 0.46 | 1.59 (1.00–2.49) | **0.047** |
| Number of pack-years smoked | | | | | | | | | |
| None | REF | | | REF | | | REF | | |
| Pack-years < 30 | 0.05 | 1.05 (0.70–1.55) | 0.814 | 0.02 | 1.02 (0.68–1.51) | 0.911 | 0.01 | 1.01 (0.67–1.51) | 0.952 |
| Pack-years > 30 | 0.40 | 1.50 (1.05–2.13) | **0.026** | 0.38 | 1.47 (1.03–2.09) | **0.035** | 0.36 | 1.44 (1.00–2.06) | **0.048** |
| Alcohol consumption | | | | | | | | | |
| None | REF | | | REF | | | REF | | |
| Alcohol <2 unit/week | −0.25 | 0.78 (0.48–1.22) | 0.286 | −0.29 | 0.75 (0.46–1.17) | 0.220 | −0.28 | 0.76 (0.46–1.20) | 0.247 |
| Alcohol 2–4 unit/week | 0.27 | 1.31 (0.90–1.87) | 0.150 | 0.24 | 1.28 (0.88–1.83) | 0.188 | 0.28 | 1.32 (0.91–1.90) | 0.139 |
| Alcohol > 4 unit/week | 0.52 | 1.69 (1.12–2.53) | **0.012** | 0.52 | 1.68 (1.11–2.51) | **0.013** | 0.50 | 1.65 (1.09–2.48) | **0.017** |

[a]BMI, body mass index; CI, confidence interval; OR, odds ratio.
[b]Adjusted by age, gender, and BMI.
[c]Adjusted by age, gender, BMI, and family history of colorectal cancer in first-degree relatives.
[d]Adjusted by age, gender, BMI, recruitment region of participants, family history of colorectal cancer in first-degree relatives, and history of colorectal polyps.
[e]Bold values indicate statistically significant difference.

characteristics, with the relative abundance of *Fusobacterium* positively correlated with the number of adenomas (Fig. 3C).

Therefore, lifestyle-associated differential genera appear to influence the development of HRA, potentially by increasing the number and size of adenomas, thereby contributing to the malignancy of colorectal polyps.

## Mediating role of microbial genera in the increased risk of high-risk adenoma due to alcohol and cigarette consumption

Since our analysis suggested significant associations between lifestyle factors and both the risk of HRA and the relative abundance of microbial genera (*Fusobacterium, Tyzzerella 4, Actinomyces, Oscillibacter*), we postulated that the lifestyle factors could influence the risk of HRA via impact on the relative abundance of lifestyle factors-associated genus. We, therefore, conducted causal mediation analyses to test this, adjusting for covariates including BMI, age, gender, family history of CRC in first-degree relatives, polyp history, recruitment region of participants, and NSAID use.

Causal mediation analyses provided evidence that the effect of alcohol consumption at the risk of HRA was partially mediated by genus *Fusobacterium* (coefficient estimates in average causal mediation effect [ACME] = 0.0072, 95% CI: 0.0005–0.0200, *P* = 0.018; coefficient estimates in average direct effect [ADE] = 0.0689, 95% CI: 0.0051–0.1400, *P* = 0.026; proportion = 9.4%) (Fig. 4A). And the effect of cigarette consumption at the risk of HRA was completely mediated by genus *Tyzzerella 4* (coefficient estimates in ACME = 0.0080, 95% CI: 0.0004–0.0200, *P* = 0.024; coefficient estimates in ADE = 0.0481, 95% CI: −0.0064 to 0.1100, *P* = 0.082; proportion = 14.2%) (Fig. 4B).

Sensitivity analyses distinguished between different outcomes, where the mediating effect of genus *Fusobacterium* and *Tyzzerella 4* remained significant when considering

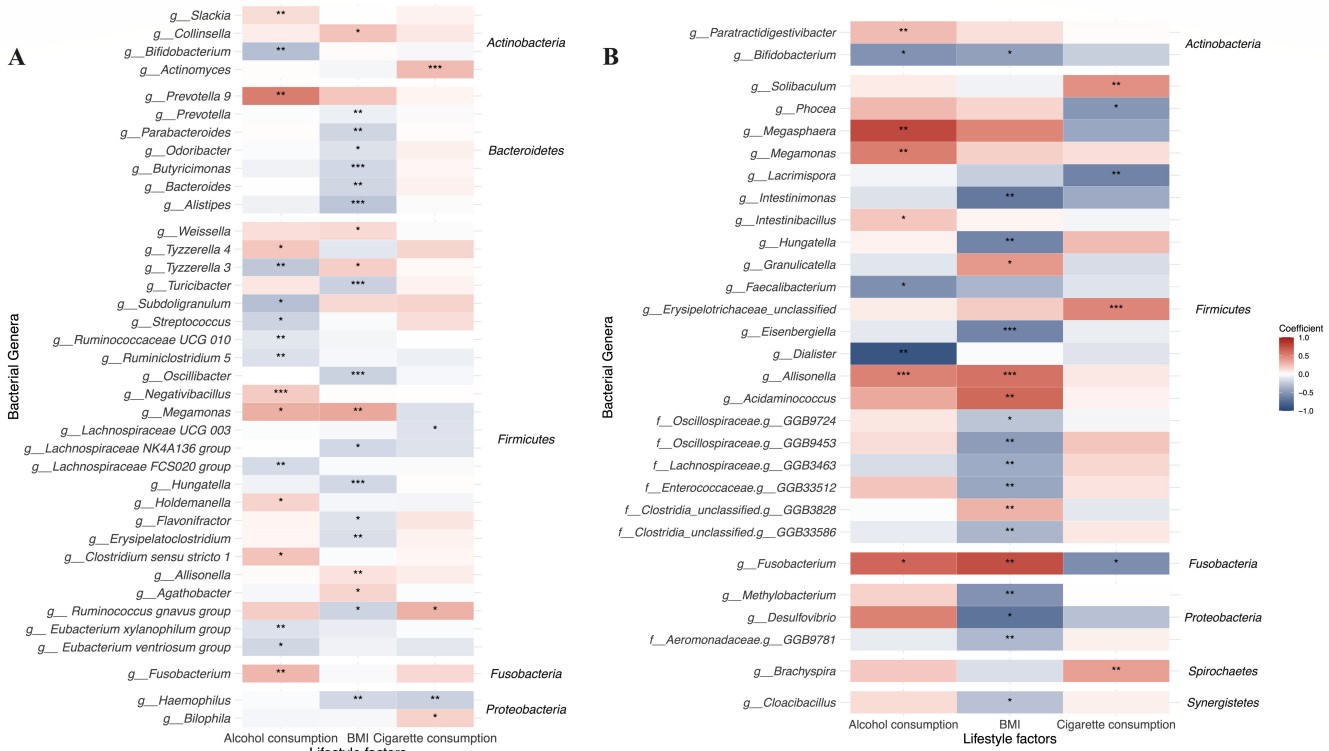

**FIG 2** Association between gut microbiota and lifestyle risk factors. (A) Heatmap of the association between gut genera and lifestyle risk factors based on the coefficients from MaAsLin2 analysis in our cohort. Genera were classified according to the phyla on the right side of the figure (false discovery rate [FDR] <0.25). (B) Heatmap of the association between gut genera and lifestyle risk factors based on the coefficients from MaAsLin2 analysis in external Japanese cohort (false discovery rate <0.25). Genera were classified according to the phyla on the right side of the figure. The asterisks in the plot indicate statistically significant differences. ***, FDR < 0.05; **, FDR < 0.01; *, FDR < 0.001. BMI, body mass index; NS, not significant; PCoA, principal coordinates analysis; PERMANOVA, permutational multivariate analysis of variance.

advanced adenoma as outcome variable, underscoring the role of genus *Fusobacterium* and *Tyzzerella 4* in mediating the impact of alcohol consumption and cigarette consumption on advanced adenomas, respectively (Fig. 4C and D).

In the subgroup analysis, the mediating effects remained in the male participants; however, when only female participants were included, we found that the mediating effect of genera *Fusobacterium* and *Tyzzerella 4* was no longer present (Fig. 4E through H). This suggests that the mediating role of these two genera in the effect of lifestyle factors on the increased risk of HRA may occur only in the male population.

## DISCUSSION

In our large-scale, multicenter CRC screening cohort, we identified key lifestyle factors— including elevated BMI, heavy smoking (>30 pack-years), and alcohol consumption (>4 units/week)—serving as independent risk factors for HRA. These modifiable exposures significantly perturbed gut microbiota profiles. Specifically, *Fusobacterium* mediated alcohol-associated HRA risk, and *Tyzzerella 4* mediated smoking-related risk, especially in male subjects. These genera were further associated with malignant features of adenomas (e.g., number and size), suggesting that the interaction between gut microbiota and lifestyle factors may influence the development of colorectal precancerous lesions.

Previous studies have established that adverse lifestyle behaviors contribute to CRC risk (30–32), but the mechanisms through which they promote carcinogenesis— especially at the precancerous stage—remain incompletely understood. Our findings help address this gap by revealing the potential role of microbiota as mediator through which lifestyle factors contribute to early colorectal neoplasia, providing mechanistic

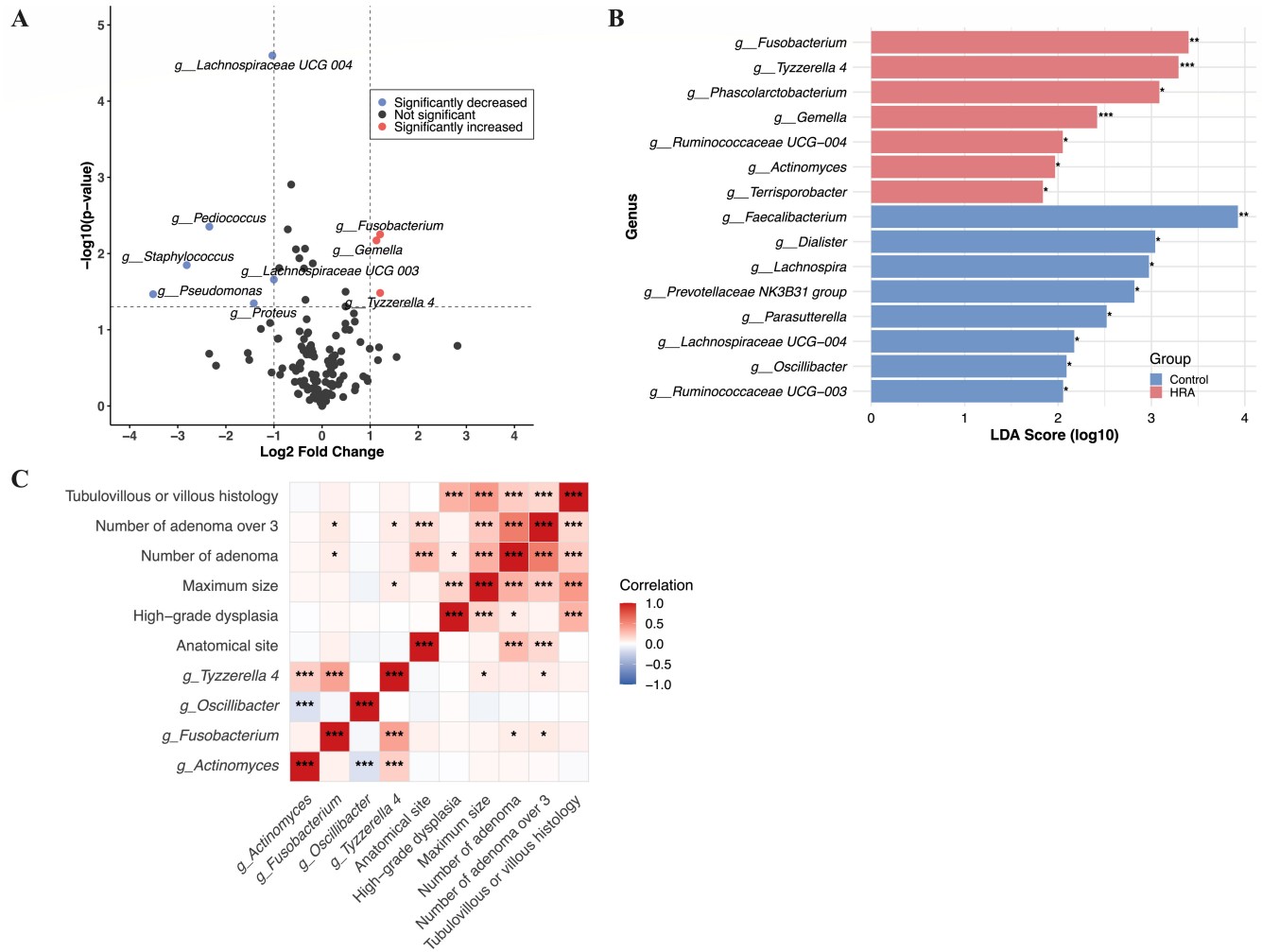

**FIG 3** Differential microbiota in the high-risk adenoma group and the correlation with adenoma characteristics. (A) Volcano plot of differences in abundance of genera between the high-risk adenoma group and the control group. Red points represent genera with increased abundance in the high-risk adenoma group, while blue points represent genera with decreased abundance. (B) LefSe analysis of the differences in genera between the high-risk adenoma group and the control group. The red bars represent genera with significantly increased abundance in the high-risk adenoma group, and the green bars represent genera with significantly increased abundance in the control group. (C) Heatmap of Spearman correlation between significantly different genera related to lifestyle factors and adenoma features.

context for epidemiological observations and highlighting new opportunities for microbiota-targeted or lifestyle-based prevention strategies.

Lifestyle factors were evaluated via patient recall of the preceding year, capturing habitual patterns. These prolonged lifestyle habits had a significant impact on gut microbiota, most notably through the alcohol-associated enrichment of *Fusobacterium*— a pattern validated in our cohort and replicated in a Japanese population (29). Among this genus, the species *Fusobacterium nucleatum* and *Fusobacterium mortiferum* have been reported to possess tumor-promoting effects in the colorectum, especially the former (33, 34). Previous studies have shown that *Fusobacterium mortiferum* is significantly elevated in the stool of patients with colorectal adenomas compared to control group (35) and is enriched in adenoma patients with APC mutations compared to those without (36). Given our demonstration of *Fusobacterium* as a mediator between lifestyle factors and the HRAs, future microbiota-based prevention strategies should prioritize targeting this genus. However, due to the limitations of 16S rRNA sequencing, we were unable to delineate microbial mediation at the species level. Thus, the specific species

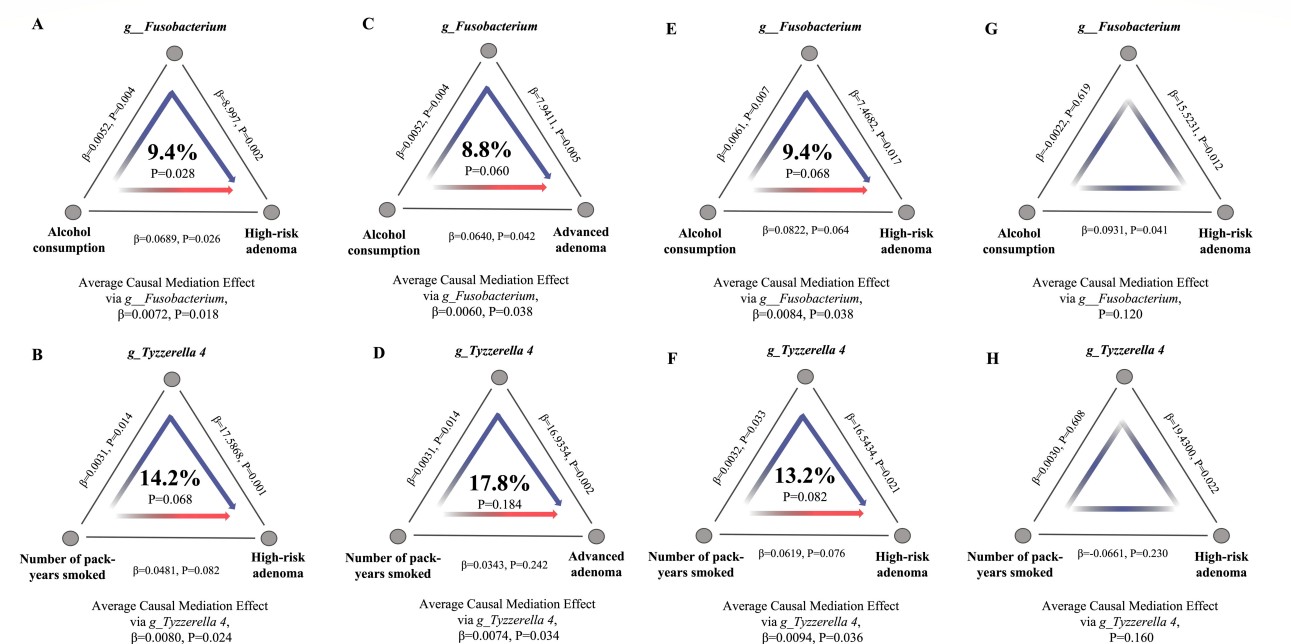

**FIG 4** The mediating role of genera in the association between lifestyle risk factors and colorectal neoplasia risk. (A) The mediating role of genus *Fusobacterium* in the association between alcohol consumption and the risk of high-risk adenoma. The middle percentage represents the mediation proportion. The β on the line segment represents the coefficient of the effect between the two variables. (B) The mediating role of genus *Tyzzerella 4* in the association between cigarette consumption and the risk of. (C) The mediating role of genus *Fusobacterium* in the association between alcohol consumption and the risk of advanced adenoma. (D) The mediating role of genus *Tyzzerella 4* in the association between cigarette consumption and the risk of advanced adenoma. (E) The mediating role of genus *Fusobacterium* in the association between alcohol consumption and the risk of high-risk adenoma in male population. (F) The mediating role of genus *Tyzzerella 4* in the association between cigarette consumption and the risk of high-risk adenoma in male population. (G) The mediating role of genus *Fusobacterium* in the association between alcohol consumption and the risk of high-risk adenoma in female population. (H) The mediating role of genus *Tyzzerella 4* in the association between cigarette consumption and the risk of high-risk adenoma in female population.

mediating alcohol-associated HRA risk remains uncharacterized and warrants future investigation.

In addition, the microbial mediation effects observed in our study appeared to be sex-specific, as significant mediation by these genera persisted exclusively in male subgroup. Accumulated evidence reveals that gut microbiota composition and function vary significantly by sex. For instance, males were shown to have lower relative abundances of *Firmicutes* and *Actinobacteria*, as well as a decreased *Firmicutes/Bacteroidetes* ratio compared to females (37). Functional differences have also been noted in an animal study by Wang et al., which demonstrated that fecal microbiota transplantation from female donors reduced systemic pro-inflammatory cytokine levels following ischemic stroke (38). Critically, sex-dependent abundance variations were observed in *Fusobacterium*, the genus of particular interest in our study, revealing potential genus-level sexual dimorphism. Similar to our findings, a fecal microbiome analysis of healthy Japanese individuals reported higher levels of *Fusobacterium* and *Prevotella* in males, while females had higher abundances of *Bifidobacterium* and *Ruminococcus* (39). Another study conducted in Chinese population found *Fusobacterium* to be more abundant in obese males, whereas obese females showed much notable enrichment in *Bifidobacterium*, *Coprococcus*, and *Dialister* (40). These consistent sexual dysmorphism suggest distinct regulatory mechanism including sexual maturation and sex hormones (41, 42), while differences in immune function may also play a role (43). Consequently, elucidating the precise biological mechanisms underlying these sex-dependent microbial patterns demands further investigation.

The strength of this study lies in our large, multicenter cohort, providing strong representation across China. Besides, we focused on the risk of HRA, a critical window for intercepting malignant transformation to CRC, revealing how lifestyle-microbiota interactions drive precancerous lesions, thereby offering valuable insights for blocking cancer progression on early stage. Moreover, the external validation and sensitivity analysis provide additional support for our conclusions. However, our study also has limitations. Our cohort was restricted to screening-age populations, compromising the generalizability across all age groups. The questionnaire-derived lifestyle data collected here, though retrospective over 12 months, remain vulnerable to recall bias. Moreover, reliance on a single assessment fails to capture exposure fluctuations, introducing potential bias. Furthermore, gut microbiota profiling was performed using 16S rRNA gene amplicon sequencing, with limited depth to fully explore species-level gut microbiota. Further exploration of the mechanistic underpinnings of the observed mediation effects is needed in future research. Additionally, although validated using a large-scale external data set, population differences between cohorts may introduce selection bias. And, both cohorts' East Asian predominance and limited lifestyle variables in public data necessitate future validation in ethnically diverse populations.

In conclusion, we integrated lifestyle factors and gut microbiome in evaluating the risk of HRAs in Chinese populations, identified that BMI, cigarette consumption, and alcohol consumption significantly increase the risk of HRA. Lifestyle factors also have a profound impact on gut microbiota, and the genus *Fusobacterium* and *Tyzzerella 4* plays a significant mediating role in the association between lifestyle risk factors and the increased risk for HRA. This study offers novel insight into the tripartite relationship among lifestyle, microbiota, and colorectal neoplasm risk, establishing a mechanistic basis for epidemiological associations and identifying potential preventive strategies through microbiota modulation or lifestyle interventions.

## ACKNOWLEDGMENTS

We sincerely appreciate all study participants and collaborating center staff for their contributions to cohort establishment. We thank Dr. Hanyuan Xu from Department of Clinical Nutrition, Peking Union Medical College Hospital, for her valuable assistance with manuscript language editing. We also acknowledge the National Genomics Data Center (China) for their support.

This work was supported by the National Key Research and Development Program of China (2024YFA0918500); Chinese Academy of Medical Science Innovation Fund for Medical Science (2022-I2M-1-0031); Beijing Research Ward Excellence Program, BRWEP (BRWEP2024W034010103); the Fundamental Research Funds for the Central Universities (3332024122); the National Natural Science Foundation of China (82173606, 82273726, 82473705); Natural Science Foundation of Beijing Municipality (7244394, 7232123); Science and Technology Projects of Xizang Autonomous Region, China (XZ202501JD0021).

D.W. and H.C. had full access to all the data in the study and takes responsibility for the integrity of the data and the accuracy of the data analysis. All the authors have read and approved the final manuscript. K.S.: Conceptualization, Data curation, Formal analysis, Funding acquisition, and Writing—original draft. J.L.: Conceptualization, Data curation, Formal analysis. Y.Z.: Data curation. D.W.: Conceptualization, Funding acquisition, and Writing—review & editing. H.C.: Conceptualization, Data curation, Funding acquisition, and Writing—review & editing. M.D.: Conceptualization, and Writing—review & editing.

## AUTHOR AFFILIATIONS

[1]Center for Prevention and Early Intervention, National Infrastructures for Translational Medicine, Institute of Clinical Medicine, Peking Union Medical College Hospital, Chinese Academy of Medical Science and Peking Union Medical College, Beijing, China

²Department of Gastroenterology, Peking Union Medical College Hospital, Chinese Academy of Medical Science and Peking Union Medical College, Beijing, China
³Department of Epidemiology, National Cancer Center/National Clinical Research Center for Cancer/Cancer Hospital, Chinese Academy of Medical Sciences and Peking Union Medical College, Beijing, China
⁴Center for Clinical and Epidemiologic Research, Beijing Anzhen Hospital, Capital Medical University, Beijing Institute of Heart, Lung and Blood Vessel Diseases, Beijing, China

## AUTHOR ORCIDs

Kai Song  http://orcid.org/0000-0003-4459-7059
Dong Wu  http://orcid.org/0000-0001-9430-9874
Hongda Chen  http://orcid.org/0000-0001-6171-1162

## AUTHOR CONTRIBUTIONS

Kai Song, Conceptualization, Data curation, Formal analysis, Funding acquisition, Writing – original draft | Jiahui Luo, Conceptualization, Data curation, Formal analysis | Dong Wu, Conceptualization, Funding acquisition, Writing – review and editing | Hongda Chen, Conceptualization, Data curation, Funding acquisition, Writing – review and editing | Min Dai, Conceptualization, Writing – review and editing.

## DATA AVAILABILITY

The raw sequence data of our cohort have been deposited in the Genome Sequence Archive in National Genomics Data Center, China National Center for Bioinformation/Beijing Institute of Genomics, Chinese Academy of Sciences (GSA: CRA028528) that are publicly accessible at https://ngdc.cncb.ac.cn/gsa. Raw sequencing data of the external cohort can be downloaded from Sequence Read Archive (SRA) using the following accession IDs: DRA006684 and DRA008156 at https://www.ncbi.nlm.nih.gov/sra.

## ETHICS APPROVAL

The study received approval from the Ethics Committee of the National Cancer Center/Cancer Hospital, Chinese Academy of Medical Sciences, and Peking Union Medical College (Approval ID: 18-013/1615) and was carried out in strict compliance with the ethical principles of the Declaration of Helsinki. Collection of all samples and documentation of clinical characteristics were conducted only after obtaining complete informed consent from each participant. We received written informed consent from all patients.

## ADDITIONAL FILES

The following material is available online.

### Supplemental Material

**Figure S1 (mSystems00933-25-s0001.tiff).** Workflow diagram for subject selection.
**Supplemental Material (mSystems00933-25-s0002.docx).** Supplemental methods, figure legend, and tables.

### Open Peer Review

**PEER REVIEW HISTORY (review-history.pdf).** An accounting of the reviewer comments and feedback.

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
