## [Reviewer comments · mSystems]

Gut microbiome mediate the associations between lifestyle factors and risk of colorectal high-risk adenoma: Results from a population-based cohort study

Kai Song, Jiahui Luo, Yuhan Zhang, Dong Wu, Hongda Chen, and Min Dai

Corresponding Author(s): Dong Wu, Peking Union Medical College Graduate School

Review Timeline:

Submission Date:	June 22, 2025
Editorial Decision:	July 24, 2025
Revision Received:	August 6, 2025
Accepted:	August 22, 2025

Editor: Marc Cook

Reviewer(s): Disclosure of reviewer identity is with reference to reviewer comments included in decision letter(s). The following individuals involved in review of your submission have agreed to reveal their identity: Jinshui Yang (Reviewer #1); Wei-Hua Chen (Reviewer #2); Zhihua Liu (Reviewer #3)

Transaction Report:

DOI: <https://doi.org/10.1128/mSystems.00933-25>

Re: mSystems00933-25 (Gut microbiomes mediate the associations between lifestyle factors and risk of colorectal high-risk adenoma: Results from a population-based cohort study)

Dear Dr. Dong Wu:

Revision Guidelines

Sincerely,
Marc Cook
Editor
mSystems

Reviewer #1 (Comments for the Author):

CRC was the third most frequently diagnosed cancer worldwide and the fourth leading cause of cancer-related deaths in China. Lifestyle choices, such as diet, smoking, and alcohol consumption, are known to influence colorectal cancer risk, but how gut bacteria contribute to this process remains unclear. This study explored the connection between these lifestyle factors, gut microbes, and colorectal precancerous growths and found that obesity, heavy smoking, and frequent alcohol linked to an

increased harmful bacteria, including *Fusobacterium* and *Tyzzarella* 4, and partially explain why smoking and alcohol raise cancer risk. The results is interesting and constructive. However, there are still some parts that need to be modified.

- 1.The title "Gut microbiomes" should be "Gut microbiome"
- 2.In line 83 to 84 Please specify which factors can increase the risk of CRC and what is the research progress?
- 3.Line 87, Suggest changing "bacterial metabolites" to "gut microbiota-derived metabolites"
- 4.Line 89, "specific genera" changed to "gut microbiota"
- 5.Line 174, "with a target q value of 0.25" Does it meet the statistical requirements? Please list the references.
- 6.Please carefully check if the Tables and figures match the description in the main text. Line 206--206 is not consistent with the Table 2. Line 286--289 is not consistent with the Figure 4A. The annotations in Figures 2 and 3 are reversed.
- 7.Line 277, "we demonstrated" change to "Our analysis suggested" would be better.
- 8.Line 227 "p value < 0.25" but in your "Materials and methods" in line 174, there is "with a target q value of 0.25." Please providing unified explanations in "Materials and methods" that The q-value represents the p-value adjusted for multiple testing using the False Discovery Rate (FDR) method.
- 9.Line 228, "applying a stricter cutoff of q < 0.10" Please indicate the basis for use and provide references.
- 10.Can the Control of the external queue represent the healthy and low-risk population in your study ?
- 11.Can the CRC population represent your high-risk group ?
- 12.Although the results show that some gut microbiota and lifestyle factors are matched in both internal and external data, significant factors such as BMI, gender, alcohol consumption, age, etc. in the internal data do not show consistent trends in the external data. Smoking alone is still significant. Can I understand that smoking alone is the main risk factor for COC ?

Reviewer #2 (Comments for the Author):

This study explored of gut microbiome-mediated mechanisms linking lifestyle to colorectal precancerous lesions. They took a multi-center effort to uniquely combines lifestyle factors (BMI, smoking, alcohol) with gut microbiome profiling to elucidate their joint impact on colorectal high-risk adenomas (HRA). This tripartite approach fills a critical gap in understanding how modifiable risk factors interact with the microbiome to drive early carcinogenesis.

Importantly, they identified *Fusobacterium* and *Tyzzarella* 4 as mediators of alcohol- and smoking-associated HRA risk, respectively, and validated their main findings in (e.g., *Fusobacterium*-alcohol link) in an external Japanese cohort enhances generalizability and strengthens the robustness of findings, a notable strength.

Minor issues:

1. a map showing the geo-locations of the centers from which the samples were collected should be used as Figure 1A, highlighting the representativeness of this study.
2. the authros should briefly discuss the novelty of this study by mentioning what is known in the field and what the results of this study could contribute.

Reviewer #3 (Comments for the Author):

This manuscript presents a comprehensive and well-designed study investigating the mediating role of gut microbiota in the association between lifestyle factors and the risk of colorectal high-risk adenomas (HRA). Leveraging a large, multicenter CRC screening cohort in China, the authors employed 16S rRNA sequencing, multivariable modeling (MaAsLin2), differential abundance analysis (LEfSe), and causal mediation analysis to explore the interplay between lifestyle exposures and gut microbial composition in relation to HRA development. It highlights high BMI, long-term smoking, and alcohol consumption are independent risk factors for HRA, as well as the potential mediating role of *Fusobacterium* and *Tyzzarella* 4. These findings emphasize the importance of lifestyle modification and microbiome-targeted strategies in the prevention of colorectal precancerous lesions and open new avenues for future mechanistic and translational research.

Major suggestions

1. The language in the manuscript needs refinement. For example, Line 265, "Combining the results from the MaAsLin 2 analysis, LEfSe analysis, we identified an overlap between the differentially abundant genera between HRA and control groups and the genera associated with lifestyle factors." It is not clear where the authors refer to. Please clarify.
2. Causal mediation analyses indicated that mediating effect of genera *Fusobacterium* and *Tyzzarella* 4 was not present in female population, it would help if the authors include the analysis in the figures. Also, it is desirable if the sex-specific effects of key genera could be discussed in the discussion part.
3. The introduction is not adequate. Expansion is needed.
4. Uploaded figures in the wrong order. Please double check.

Response to reviewers

We would like to sincerely thank the reviewers for their thorough evaluation and constructive feedback, which have been instrumental in improving this manuscript. The reviewers' insights, addressing both the methodology and presentation of our work, have significantly enhanced the scientific rigor and clarity of the paper. The process of addressing these comments has provided us with a valuable opportunity to refine our analysis and elevate the overall quality of the manuscript. We also extend our appreciation to the editorial team for facilitating this valuable review process.

Data Availability Statement:

We gratefully acknowledge the editorial team's valuable support during the review process. The raw sequencing data from this study have been deposited in the Genome Sequence Archive (GSA) at the National Genomics Data Center, China National Center for Bioinformation / Beijing Institute of Genomics, Chinese Academy of Sciences (accession number: CRA028528) and are publicly available at <https://ngdc.cncb.ac.cn/gsa>, as specified in our Data Availability section.

The dataset will be made publicly available either upon publication of our first article cited this accession number(i.e., the present study) or by July 30, 2027, whichever occurs first.

Reviewer #1 (Comments for the Author):

CRC was the third most frequently diagnosed cancer worldwide and the fourth leading cause of cancer-related deaths in China. Lifestyle choices, such as diet,

smoking, and alcohol consumption, are known to influence colorectal cancer risk, but how gut bacteria contribute to this process remains unclear. This study explored the connection between these lifestyle factors, gut microbes, and colorectal precancerous growths and found that obesity, heavy smoking, and frequent alcohol linked to an increased harmful bacteria, including Fusobacterium and Tyzzerella 4, and partially explain why smoking and alcohol raise cancer risk. The results is interesting and constructive. However, there are still some parts that need to be modified.

1.The title "Gut microbiomes" should be "Gut microbiome"

Response: We sincerely appreciate your kind reminder. The suggested revisions have been made.

2.In line 83 to 84 Please specify which factors can increase the risk of CRC and what is the research progress?

Response: Thank you for your valuable suggestion. We have revised the relevant section as follows:

“While multiple studies in CRC have shown that unhealthy lifestyle factors, including excessive intake of red meat, smoking, and alcohol consumption, increase disease risk, evidence clarifying their contribution to HRA remains scarce. Moreover, the underlying mechanism through which these external exposures promote neoplastic transformation at the precancerous stage are not fully understood.

The gut microbiota, as a key component of the intestinal microecosystem interacting with the gut epithelial mucosa,is closely linked to colorectal adenoma

and CRC development through compositional alterations. These alterations, including the production of metabolites that induce chronic inflammation and immune dysfunction, as well as elevated serum lipopolysaccharide levels, suggest that the gut microbiota may mediate the impact of adverse lifestyle factors on colorectal neoplastic risk, warranting further investigation. However, while this overall mediating role is plausible, evidence at the taxonomic level remains scarce. Specifically, the gut microbial taxa that interact with particular lifestyle factors to modulate HRA risk have yet to be clearly identified.”.

3.Line 87, Suggest changing "bacterial metabolites" to "gut microbiota-derived metabolites"

Response: Thanks for your kind reminder. The revisions have been made accordingly.

4.Line 89, "specific genera" changed to "gut microbiota"

Response: Thanks for your kind reminder. The revisions have been made accordingly.

5.Line 174, "with a target q value of 0.25" Does it meet the statistical requirements? Please list the references.

Response: Thank you for your thoughtful comments regarding our methodological approach. The use of a 0.25 threshold was based on the recommendation of the authors who introduced the MaAsLin2 method (PLoS Comput Biol. 2021; 17:e1009442). In addition, we referred to several high-quality microbiome studies that adopted similar significance thresholds (Nat Med. 2021; 27:333–43; Nat

Commun. 2021; 12:4845; mSystems. 2022; 7:e0000422). We have added these references to the Methods section in the revised manuscript.

6. Please carefully check if the Tables and figures match the description in the main text. Line 206--206 is not consistent with the Table 2. Line 286--289 is not consistent with the Figure 4A. The annotations in Figures 2 and 3 are reversed.

Response: We sincerely apologize for the confusion caused by our mistake. Due to multiple rounds of internal revisions during the preparation of the manuscript, this part of the data was not updated in accordance with the latest results. We have now carefully reviewed and revised the relevant sections to ensure consistency and accuracy. Additionally, the legends for Figures 2 and 3 were incorrect due to an error in the upload sequence during submission, and have been corrected accordingly.

7. Line 277, "we demonstrated" change to "Our analysis suggested" would be better.

Response: We sincerely appreciate your constructive suggestion. We agree with your perspective and have modified the expression from "we demonstrated" to "Our analysis suggested".

8. Line 227 "p value < 0.25" but in your "Materials and methods" in line 174, there is "with a target q value of 0.25." Please providing unified explanations

in "Materials and methods" that The q-value represents the p-value adjusted for multiple testing using the False Discovery Rate (FDR) method.

Response: We sincerely appreciate your rigorous and careful review. Our original intention was to indicate that the q value represents the p value adjusted for multiple testing using the Benjamini-Hochberg method. However, to avoid redundancy in expression, we have removed the description of the q value and now refer uniformly to FDR. In the Methods section, we have revised the corresponding description as follows:

"Multiple hypothesis testing was addressed by controlling the false discovery rate (FDR < 0.25) via the Benjamini-Hochberg method."

9. Line 228, "applying a stricter cutoff of $q < 0.10$ " Please indicate the basis for use and provide references.

Response: We fully agree with your comment. The use of FDR < 0.1 was a subjective decision, and to improve clarity and consistency, we have removed this criterion from the revised manuscript and retained only the commonly accepted threshold of FDR < 0.25.

10.Can the Control of the external queue represent the healthy and low-risk population in your study?

Response: Thanks for your rigorous and professional comments regarding the methodology. Your concerns are very reasonable and likely stem from insufficient explanation of our study design and the purpose of using the external dataset. We have revised the Methods section to clarify this point as follows: "In addition to

conducting correlation analysis in our own dataset, an external cohort of the Japanese population was used to validate the association between lifestyle factors and gut microbial genera(details about the external cohort see supplementary material) ". One of the main goals of this study is to investigate the impact of lifestyle factors on the gut microbiota. Therefore, the inclusion of the Japanese cohort primarily serves as an external validation based on a cohort completely independent from ours, aiming to further confirm relatively robust and significant associations between lifestyle and gut microbiota. And we adjusted for covariates including disease status, age, and gender in the external cohort analysis using the MaAsLin2 method, to reduce the potential confounding effect of disease status and other covariates on the gut microbiota. Therefore, although the control group in the Japanese cohort includes only healthy controls—unlike our internal cohort, which includes both healthy controls and low-risk adenoma patients—adjusting for disease status allowed us to minimize confounding effects and maximize the utility of this public dataset for external validation. This approach strengthens the reliability of certain gut microbiota–lifestyle associations observed in our study.

However, your concern is indeed valid. Differences in population characteristics between the Japanese cohort and our internal cohort may introduce selection bias. Although we attempted to mitigate the impact of disease status on the lifestyle–microbiota associations by adjusting for disease status, this approach cannot completely eliminate confounding effects, which represents one limitation of our study. Therefore, we have added the following statement in the Discussion section to address this limitation:

"Additionally, although validated using a large-scale external dataset, population differences between cohorts may introduce selection bias. And, both cohorts' East

Asian predominance and limited lifestyle variables in public data necessitate future validation in ethnically diverse populations."

11.Can the CRC population represent your high-risk group?

Response: Thanks for your rigorous and thoughtful comments. We believe this concern is similar to point 10. Although CRC represents a more advanced colorectal neoplastic state, we adjusted for disease status in the MaAsLin2 analysis to minimize its potential confounding effect on the association between gut microbiota and lifestyle factors.

12.Although the results show that some gut microbiota and lifestyle factors are matched in both internal and external data, significant factors such as BMI, gender, alcohol consumption, age, etc. in the internal data do not show consistent trends in the external data. Smoking alone is still significant. Can I understand that smoking alone is the main risk factor for CRC?

Response: Thanks for your insightful comments. Although BMI, gender, alcohol consumption, age and cigarette consumption showed significant differences between two groups in our cohort—and BMI , cigarette consumption and alcohol consumption were identified as independent risk factors for high-risk adenomas in our cohort—such differences were not observed in the Japanese population, possibly due to population heterogeneity. However, this does not affect the primary purpose of using this external dataset, which was to investigate the associations between lifestyle factors and gut microbiota. And due to this purpose, we mainly searched publicly available datasets based on the availability of lifestyle-related variables. Although there are many publicly available colorectal neoplasm-related

datasets (e.g., those from the Sequence Read Archive and European Nucleotide Archive, including China (PRJEB10878), India (PRJNA531273), the United States (PRJEB12449), Austria (ERP008729), Italy (PRJNA447983), France (ERP008729), and Germany (PRJEB27928)), only the Japanese dataset contains the key lifestyle variables we focus on, such as smoking and alcohol consumption. Therefore, we used this dataset for external validation.

Meanwhile, lifestyle factors such as BMI, alcohol consumption, and smoking, which we focus on, have been widely recognized as risk factors for colorectal cancer (CRC) across diverse populations and studies. Our study specifically focuses on the risk impact of these lifestyle factors on precancerous lesions and innovatively explores the mediating role of gut microbiota, thereby providing potential new targets for intervention or non-invasive detection to elucidate mechanisms by which adverse lifestyles increase CRC risk.

Regarding your comment on validating CRC-related risk factors, we fully appreciate your emphasis on external validation. However, currently available public datasets do not allow identification of high-risk adenomas within adenoma populations due to the lack of detailed adenoma characteristics for individual patients. Therefore, we could not further validate lifestyle-related risk factors for high-risk adenomas in external datasets. Nevertheless, our cohort, drawn from a large multicenter Chinese population with a substantial number of patients, is representative, and the high-risk adenoma-related risk factors we identified are consistent with prior CRC studies, which supports the reliability of our conclusions.

Reviewer #2 (Comments for the Author):

This study explored of gut microbiome-mediated mechanisms linking lifestyle to

colorectal precancerous lesions. They took a multi-center effort to uniquely combines lifestyle factors (BMI, smoking, alcohol) with gut microbiome profiling to elucidate their joint impact on colorectal high-risk adenomas (HRA). This tripartite approach fills a critical gap in understanding how modifiable risk factors interact with the microbiome to drive early carcinogenesis.

Importantly, they identified *Fusobacterium* and *Tyzzarella* 4 as mediators of alcohol- and smoking-associated HRA risk, respectively, and validated their main findings in (e.g., *Fusobacterium*-alcohol link) in an external Japanese cohort enhances generalizability and strengthens the robustness of findings, a notable strength.

Minor issues:

1. a map showing the geo-locations of the centers from which the samples were collected should be used as Figure 1A, highlighting the representativeness of this study.

Response: Thank you for your kind reminder. Following your suggestion, we have added a schematic diagram showing the geographic locations of each center.

2. the authors should briefly discuss the novelty of this study by mentioning what is known in the field and what the results of this study could contribute.

Response: Thanks for your constructive comments. We have revised and added corresponding descriptions in the Discussion section, as follows:

“In our large-scale, multicenter CRC screening cohort, we identified key lifestyle factors—including elevated BMI, heavy smoking (> 30 pack-years), and alcohol consumption (> 4 units/week)—serve as independent risk factors for HRA. These modifiable exposures significantly perturbed gut microbiota profiles. Specifically, *Fusobacterium* mediated alcohol-associated HRA risk, and *Tyzzarella*

4 mediated smoking-related risk, especially in male subjects. These genera were further associated with malignant features of adenomas (e.g., number and size), suggesting that the interaction between gut microbiota and lifestyle factors may influence the development of colorectal precancerous lesions.

Previous studies have established that adverse lifestyle behaviors contribute to CRC risk, but the mechanisms through which they promote carcinogenesis—especially at the precancerous stage—remain incompletely understood. Our findings help address this gap by revealing the potential role of microbiota as mediator through which lifestyle factors contribute to early colorectal neoplasia, providing mechanistic context for epidemiological observations and highlighting new opportunities for microbiota-targeted or lifestyle-based prevention strategies.”.

Reviewer #3 (Comments for the Author):

This manuscript presents a comprehensive and well-designed study investigating the mediating role of gut microbiota in the association between lifestyle factors and the risk of colorectal high-risk adenomas (HRA). Leveraging a large, multicenter CRC screening cohort in China, the authors employed 16S rRNA sequencing, multivariable modeling (MaAsLin2), differential abundance analysis (LEfSe), and causal mediation analysis to explore the interplay between lifestyle exposures and gut microbial composition in relation to HRA development. It highlights high BMI, long-term smoking, and alcohol consumption are independent risk factors for HRA, as well as the potential mediating role of *Fusobacterium* and *Tyzzarella* 4. These findings emphasize the importance of lifestyle modification and microbiome-targeted strategies in the prevention of colorectal precancerous lesions and open new avenues for future mechanistic and translational research.

Major suggestions

1. The language in the manuscript needs refinement. For example, Line 265,

"Combining the results from the MaAsLin 2 analysis, LEfSe analysis, we identified an overlap between the differentially abundant genera between HRA and control groups and the genera associated with lifestyle factors." It is not clear where the authors refer to. Please clarify.

Response: Thank you for this valuable comment. After thorough revision and language editing of the manuscript, we have updated the sentence as follows: 'Integrating findings from both MaAsLin 2 and LEfSe analyses revealed significant overlap between genera differentially abundant in HRA versus controls and those associated with lifestyle factors.'

2. Causal mediation analyses indicated that mediating effect of genera Fusobacterium and Tyzzerella 4 was not present in female population, it would help if the authors include the analysis in the figures. Also, it is desirable if the sex-specific effects of key genera could be discussed in the discussion part.

Response: Thank you for your kind reminder and constructive suggestions. We have added the results for the female subgroup in the figure. It can be observed that in the female subgroup, the associations between lifestyle factors and the gut microbiota disappeared, and the mediating effects became non-significant ($P > 0.05$). In addition, we have added a description in the Discussion section regarding the sex-specific effects of the gut microbiota, as follows:

“In addition, the microbial mediation effects observed in our study appeared to be sex-specific, as significant mediation by these genera persisted exclusively in male subgroup. Accumulated evidence reveals that gut microbiota composition and function vary significantly by sex. For instance, males were shown to have lower

relative abundances of *Firmicutes* and *Actinobacteria*, as well as an decreased *Firmicutes/Bacteroidetes* ratio compared to females. Functional differences have also been noted in an animal study by Wang et al., which demonstrated that fecal microbiota transplantation from female donors reduced systemic pro-inflammatory cytokine levels following ischemic stroke. Critically, sex-dependent abundance variations were observed in *Fusobacterium*, the genus of particular interest in our study, revealing potential genus-level sexual dimorphism. Similar to our findings, a fecal microbiome analysis of healthy Japanese individuals reported higher levels of *Fusobacterium* and *Prevotella* in males, while females had higher abundances of *Bifidobacterium* and *Ruminococcus*. Another study conducted in Chinese population found *Fusobacterium* to be more abundant in obese males, whereas obese females showed much notable enrichment in *Bifidobacterium*, *Coprococcus*, and *Dialister*. These consistent sexual dimorphism suggest distinct regulatory mechanism including sexual maturation and sex hormones, while differences in immune function may also play a role. Consequently, elucidating the precise biological mechanisms underlying these sex-dependent microbial patterns demands further investigation.”

3. The introduction is not adequate. Expansion is needed.

Response: We sincerely appreciate your constructive suggestions. We have comprehensively restructured and revised the Introduction section to better frame our study's rationale and objectives. The updated version now provides: (1) clearer theoretical background, (2) more focused research gaps, and (3) enhanced logical flow connecting existing knowledge to our investigation:

“Colorectal cancer (CRC) was the third most frequently diagnosed cancer worldwide in 2022, accounting for 9.6% of all new cancer cases and causing a

significant health and socioeconomic burden[1]. In China, CRC is the second most common cancer and the fourth leading cause of cancer mortality[2]. Prevention and early intervention targeting precancerous lesions is a more cost-effective strategy for reducing the incidence and mortality of CRC[3, 4]. Colorectal high-risk adenomas (HRAs) are a category of precancerous lesions defined based on clinical research evidence over the past decade, including advanced adenoma and cases with 3 or more adenomas[5]. Despite resection, these lesions remain at high risk for both recurrence and malignant transformation to CRC, indicating inherent aggressive potential[5, 6]. Therefore, understanding the risk factors and underlying mechanisms contributing to HRA development is essential for improving early prevention strategies.

While multiple studies in CRC have shown that unhealthy lifestyle factors, including excessive intake of red meat[7, 8], smoking[9], and alcohol consumption[9], increase disease risk, evidence clarifying their contribution to HRA remains scarce. Moreover, the underlying mechanism through which these external exposures promote neoplastic transformation at the precancerous stage are not fully understood.

The gut microbiota, as a key component of the intestinal microecosystem interacting with the gut epithelial mucosa, is closely linked to colorectal adenoma and CRC development through compositional alterations[10-13]. These alterations, including the gut microbiota-derived metabolites that induce chronic inflammation and immune dysfunction, as well as elevated serum lipopolysaccharide levels[14], suggest that the gut microbiota may mediate the impact of adverse lifestyle factors on colorectal neoplastic risk, warranting further investigation. However, while this overall mediating role is plausible, evidence at the taxonomic level remains

scarce. Specifically, the gut microbial taxa that interact with particular lifestyle factors to modulate HRA risk have yet to be clearly identified.

To address this gap, we conducted a large, multicenter cohort study recruiting participants undergoing CRC screening across China. Our aim was to identify lifestyle-related risk factors for HRA, evaluate their associations with gut microbiota, and identify certain bacterial taxa mediating these relationships. This study provides a novel perspective on the microbial mechanisms linking lifestyle and early colorectal neoplasia, contributing to establishing microbiota-based or lifestyle-targeted strategies in preventing CRC from its precancerous stage.”

4. Uploaded figures in the wrong order. Please double check.

Response: We sincerely apologize for the confusion caused by our mistake. The legends for Figures 2 and 3 were incorrect due to an error in the upload sequence during submission, and have been corrected accordingly.

Re: mSystems00933-25R1 (Gut microbiome mediate the associations between lifestyle factors and risk of colorectal high-risk adenoma: Results from a population-based cohort study)

Dear Dr. Dong Wu:

Your manuscript has been accepted, and I am forwarding it to the ASM production staff for publication. Your paper will first be checked to make sure all elements meet the technical requirements. ASM staff will contact you if anything needs to be revised before copyediting and production can begin. Otherwise, you will be notified when your proofs are ready to be viewed.

Cover Image Submissions: If you would like to submit a potential Cover Image, please email a file and a short legend to mSystems@asmusa.org. Please note that we can only consider images that (i) the authors created or own and (ii) have not been previously published. By submitting, you agree that the image can be used under the same terms as the published article. Image File requirements: TIF/EPS, 7.5 inches wide by 8.25 inches tall (at least 2,250 pixels wide by 2,475 pixels tall), minimum 300 dpi resolution (600 dpi preferred), RGB, and no figure elements, e.g., arrows or panel labels. The legend should be a short description of the image, 1-2 sentences recommended. Please download and use this interactive template in Adobe to ensure that your proposed cover image meets our size requirements (<https://journals.asm.org/pb-assets/pdf-text-excel-files/ASM-Interactive-Sizing-Cover-Template-1715689791.pdf>).

Sincerely,
Marc Cook
Editor
mSystems

Reviewer #1 (Comments for the Author):

CRC was the third most frequently diagnosed cancer worldwide and the fourth leading cause of cancer-related deaths in China. This study explored the connection between these lifestyle factors, gut microbes, and colorectal precancerous growths and found that obesity, heavy smoking, and frequent alcohol linked to an increased harmful bacteria, including Fusobacterium and Tyzzerella 4, and partially explain why smoking and alcohol raise cancer risk. The results is interesting and constructive.

Reviewer #2 (Comments for the Author):

all my previous concerns have been sufficient addressed. I have no further comments.